# Estradiol-induced immune suppression via prostaglandin E$_2$ during parturition in bovine leukemia virus-infected cattle

Yamato Sajiki[1], Satoru Konnai[1,2]*, Tomohiro Okagawa[2], Naoya Maekawa[2], Shinya Goto[1], Junko Kohara[3], Atsushi Nitanai[4], Hirofumi Takahashi[4], Kentaro Kubota[5], Hiroshi Takeda[6], Shiro Murata[1,2], Kazuhiko Ohashi[1,2]

1 Department of Disease Control, Faculty of Veterinary Medicine, Hokkaido University, Sapporo, Japan,
2 Department of Advanced Pharmaceutics, Faculty of Veterinary Medicine, Hokkaido University, Sapporo, Japan, 3 Animal Research Center, Agriculture Research Department, Hokkaido Research Organization, Shintoku, Japan, 4 Hokkaido Higashi Agricultural Mutual Aid Association, Nakashibetsu, Japan, 5 Rumoi Nambu Veterinary Clinic, Hokkaido Chuo Agricultural Mutual Aid Association, Tomamae, Japan, 6 Osyamanbe Veterinary Clinic, Minami Hokkaido Agricultural Mutual Aid Association, Osyamanbe, Japan

* konnai@vetmed.hokudai.ac.jp

**Data Availability Statement:** All relevant data are within the paper and its Supporting Information files.

## Abstract

Immune suppression during pregnancy and parturition is considered a risk factor that is related to the progression of bovine chronic diseases, such as bovine leukosis, which is caused by bovine leukemia virus (BLV). Our previous studies have demonstrated that prostaglandin E$_2$ (PGE$_2$) suppresses BLV-specific Th1 responses and contributes to the disease progression during BLV infection. Although PGE$_2$ reportedly plays important roles in the induction of parturition, PGE$_2$ involvement in immune suppression during parturition is unknown. To investigate its involvement, we analyzed PGE$_2$ kinetics and Th1 responses in BLV-infected pregnant cattle. PGE$_2$ concentrations in sera were increased, whereas IFN-γ responses were decreased before delivery. PGE$_2$ is known to suppress Th1 immune responses in cattle. Thus, these data suggest that PGE$_2$ upregulation inhibits Th1 responses during parturition. We also found that estradiol was important for PGE$_2$ induction in pregnant cattle. *In vitro* analyses indicated that estradiol suppressed IFN-γ production, at least in part, via PGE$_2$/EP4 signaling. *In vivo* analyses showed that estradiol administration significantly influenced the induction of PGE$_2$ production and impaired Th1 responses. Our data suggest that estradiol-induced PGE$_2$ is involved in the suppression of Th1 responses during pregnancy and parturition in cattle, which could contribute to the progression of BLV infection.

## Introduction

In dairy cattle, efficient milk production continues to require for pregnancy and 3 parturition each year. During pregnancy and parturition, maternal immune responses are suppressed to contribute to fetal survival, which is also known as maternal tolerance of the fetus [1, 2]. In general, Th1/Th2 imbalance and regulatory T cell (Treg) induction are known to be involved

**Funding:** This work was supported by JSPS KAKENHI grant number 19KK0172 [to S.K.], grants from the Project of the NARO, Bio-oriented Technology Research Advancement Institution (Research Program on Development of Innovative Technology 26058 BC [to S.K.] and Special Scheme Project on Regional Developing Strategy, Grant 16817557 [to S.K.]), Regulatory research projects for food safety, animal health and plant protection (JPJ008617.17935709) funded by the Ministry of Agriculture, Forestry and Fisheries of Japan and Clinical Research Promotion Fund 2021 by Hokkaido University Veterinary Teaching Hospital[to S.K.]. The funders had no role in study design, data collection and analysis, decision to publish, or preparation of the manuscript.

**Competing interests:** The authors have declared that no competing interests exist.

in maternal tolerance [3, 4]. Sex hormones, such as estradiol and progesterone, play important roles in the inhibition of Th1 responses and induction of Tregs [5, 6]. Previous report has shown that sex hormones contribute to the suppression of immune responses in pregnant cattle [7]. Although immune suppression in pregnant animals is essential for the prevention of fetal injury by the maternal immune system, this suppression may affect the progression and onset of bovine chronic diseases, such as bovine leukosis. Bovine leukemia virus (BLV) is a member of the genus *Deltaretrovirus*, and causes bovine leukosis after a long latent period [8]. Most BLV-infected cattle remain subclinical and are referred to as aleukemic. However, approximately 30% of BLV-infected cattle develop persistent lymphocytosis, which is characterized by the nonmalignant polyclonal expansion of infected B cells in peripheral blood. Less than 5% of BLV-infected cattle develop bovine leukosis, which is called enzootic bovine leukosis (EBL) and is characterized by fatal lymphoma or lymphosarcoma [9]. The suppression of the immune system, particularly in a stressful situation such as parturition, is considered a risk factor for the development of bovine leukosis [10, 11]. Although parturition is an important event for progression to the clinical stages of BLV infection, the relationship between parturition and the progression of these diseases has not been fully elucidated.

Prostaglandin $E_2$ ($PGE_2$) is an inflammatory mediator that is metabolically derived from arachidonic acid by cyclooxygenase enzymes (COX-1 and COX-2) [12]. COX-1 is a constitutive enzyme that is expressed in many tissues including the kidneys, stomach, and vascular endothelium, and it is involved in a multitude of physiological processes [13]. Conversely, COX-2 is an inducible enzyme that is regulated by inflammatory cytokines, antigen-stimulation, and growth factors via the activation of nuclear factor-kappa B (NF-κB) [14]. $PGE_2$ inhibits the activities of immune cells such as T cells, dendritic cells, and natural killer cells via prostaglandin E receptor (EP) 2 and EP4, thereby contributing to immune evasion during chronic infections [15]. Our previous studies have demonstrated that $PGE_2$ inhibits Th1 responses in cattle, and is associated with the progression of BLV infection [16, 17]. In addition, $PGE_2$, as well as $PGF_{2\alpha}$, is an important factor related to the induction of parturition in cattle [18]. However, little information is available regarding the immune dysfunctional effect of $PGE_2$ in periparturient period.

Here, to investigate whether $PGE_2$ is involved in the suppression of Th1 immune responses in the periparturient period, $PGE_2$ kinetics and Th1 cytokine production were analyzed by using BLV-infected pregnant cattle. Additionally, to examine whether $PGE_2$-induced suppression of Th1 response is mediated by estradiol, the effects of estradiol on $PGE_2$ production and Th1 responses was analyzed by using BLV-infected and uninfected cattle.

## Materials and methods

### Animals and blood collection

The blood samples of BLV-infected and BLV-uninfected cattle (adult female, Holstein breed), which were used in this study, were obtained from several farmers and veterinarians in Japan. BLV infection was confirmed by detection of the provirus using nested polymerase chain reaction (PCR) that targets the viral LTR and by detection of the anti-BLV antibody using a commercial ELISA kit (JNC, Tokyo, Japan), as described previously [19]. The blood samples of BLV-infected pregnant cattle (animals #1–#5, Table 1) were collected at two points before delivery, on the day of delivery (day 0), and on three points after delivery (days 1, 7, and 14). Blood collection on day 0 was performed before parturition. These animal experiments were approved by the Ethics Committee of the Faculty of Veterinary Medicine, Hokkaido University. Verbal informed consent was obtained from the owners for the participation of their animals in this study.

**Table 1. BLV-infected cattle used in this study.**

| Cattle | #1 | #2 | #3 | #4 |
|---|---|---|---|---|
| Age | 98 months old | 41 months old | 56 months old | 51 months old |
| Breed | Holstein | Holstein | Holstein | Holstein |
| Sex | female | female | female | female |
| BLV infection | + | + | + | + |
| Pregnancy | + | + | + | + |
| Cattle | #5 | #6 | #7 | |
| Age | 52 months old | 98 months old | 75 months old | |
| Breed | Holstein | Holstein | Holstein | |
| Sex | female | female | female | |
| BLV infection | + | + | + | |
| Pregnancy | + | - | - | |
| Cattle | #8 | #9 | #10 | |
| Age | 85 months old | 106 months old | 152 months old | |
| Breed | Holstein | Holstein | Holstein | |
| Sex | female | female | female | |
| BLV infection | + | + | + | |
| Pregnancy | - | - | - | |

For estradiol administration to BLV-infected cattle, 5 BLV-infected cattle (animals #6–#10, Table 1) were kept in an animal facility at the Faculty of Veterinary Medicine, Hokkaido University (Sapporo, Hokkaido, Japan, animals #6 and #7), and a biosafety level I animal facility at the Animal Research Center, Agricultural Research Department, Hokkaido Research Organization (Shintoku, Hokkaido, Japan, animals #8–#10). These animal experiments were approved by the Ethics Committee of the Faculty of Veterinary Medicine, Hokkaido University, and the Ethics Committee of the Animal Research Center, Agricultural Research Department, Hokkaido Research Organization.

Blood samples were shipped immediately after collection and stored at 4˚C until the experiment. After 24 hours from collection, blood samples were used for the following analyses.

### Specimens of cattle with bovine leukosis

In this study, we collected clinical data of 68 cattle which were diagnosed with bovine leukosis in a specific area in Hokkaido, Japan from 2016 to 2018, and analyzed the time intervals between parturition and the onset of bovine leukosis. Animals with lymphoma were clinically diagnosed as onset. The diagnosis was carried out by clinical veterinarians in Livestock Hygiene Service Center, Meat Inspection Center or Agricultural Mutual Aid Association in Hokkaido, Japan.

### Preparation of peripheral blood mononuclear cells (PBMCs)

Buffy coat fraction was separated from blood samples by centrifugation (1,700 × g, 15 min, 25˚C, without break). Next, PBMCs were purified from the buffy coat fraction by density gradient centrifugation (1,200 × g, 20 min, 25˚C, without break) on 60% Percoll (GE Healthcare, Little Chalfont, UK). Then, collected PBMCs were washed 3 times by centrifugation (770 × g, 10 min, 25˚C) in phosphate buffered saline and filtered through a 40-μm cell strainer (BD Biosciences, San Jose, CA, USA). Then, PBMCs were stained with 0.4% Trypan Blue Stain

(Thermo Fisher Scientific, Waltham, MA, USA) and the number of the viable cells was counted using Countess II FL Automated Cell Counter (Thermo Fisher Scientific).

## Whole-blood culture

Whole blood cells (500 μL of blood) were incubated with 20 μg/ml concanavalin A (Con A, Sigma-Aldrich, St. Louis, MO, USA) at 37˚C under 5% $CO_2$ for 24 hours using 48-well plates (Corning Inc., Corning, NY, USA). Three culture wells were prepared in each sample. Collected culture supernatants were assayed for IFN-γ by Bovine IFN-γ ELISA Development Kit (Mabtech, Nacka Strand, Sweden) according to the manufacturer's instructions.

## Real-time PCR

Total RNA of PBMCs was extracted using TRI Reagent (Molecular Research Center, Cincinnati, OH, USA) in accordance with the manufacturer's instructions. cDNA was synthesized from the total RNA by using PrimeScript Reverse Transcriptase (Takara Bio, Otsu, Japan) according to the manufacturer's instructions. To confirm the mRNA expression of *COX2* in PBMCs, real-time PCR was performed in triplicate with the LightCycler 480 System II (Roche Diagnostic, Mannheim, Germany) using SYBR Premix DimerEraser (Takara Bio) following the manufacturer's instructions. *β-actin* (*ACTB*) was used as a reference gene. The relative expression levels were calculated by using the ΔΔCt method, and the results were indicated as relative change to no treatment group. Primers were `5'-ACG TTT TCT CGT GAA GCC CT-3'` and `5'-TCT ACC AGA AGG GCG GGA TA-3'` for *COX2*, and `5'-TCT TCC AGC CTT CCT TCC TG-3'` and `5'-ACC GTG TTG GCG TAG AGG TC-3'` for *ACTB*.

## Quantitation of PGE₂ and estradiol by ELISA

$PGE_2$ concentrations in sera were determined using Prostaglandin $E_2$ Express ELISA Kit (Cayman Chemical, Ann Arbor, MI, USA) in accordance with the previous report [16]. Estradiol concentrations in sera were determined using Estradiol ELISA Kit (Cayman Chemical) following the manufacturer's instructions. The measurement of both $PGE_2$ and estradiol was performed in triplicate.

## PBMC culture assays

PBMCs were cultured in 200 μL of RPMI 1640 medium (Sigma-Aldrich) containing 10% heat-inactivated fetal calf serum (Thermo Fisher Scientific), 100 U/ml penicillin (Thermo Fisher Scientific), 100 μg/ml streptomycin (Thermo Fisher Scientific), and 2 mM L-glutamine (Thermo Fisher Scientific). 96-well Plates (Corning Inc.) were used for all of the PBMC cultures.

To evaluate the immunosuppressive functions of estradiol *in vitro*, PBMCs ($1 \times 10^6$ cells/well) from uninfected and BLV-infected cattle (*n* = 6 each) were incubated with 2 μg/ml of 17β-Estradiol (Cayman Chemical) and optimal antigen stimulations as shown below. Dimethyl sulfoxide (DMSO, Nacalai Tesque Inc., Kyoto, Japan) was used as a vehicle of 17β-Estradiol. PBMCs from BLV-uninfected cattle were cultured in the presence of 1 μg/ml anti-CD3 monoclonal antibody (mAb) (MM1A; WSU Monoclonal Antibody Center, Pullman, WA, USA) and 1 μg/ml anti-CD28 mAb (CC220; Bio-Rad, Hercules, CA, USA) for 24 hours. PBMCs from BLV-infected cattle were cultured in the presence of fetal lamb kidney (FLK)-BLV antigen (2% heat-inactivated culture supernatant of FLK-BLV cells) for 6 days. As described in a previous paper, FLK-BLV supernatant contained BLV antigens to activate BLV-

specific T cells in PBMC cultures [20]. After incubation, culture supernatants were collected and IFN-γ concentrations were quantitated by ELISA, as described above.

To examine the effects of estradiol on the gene expression of *COX2*, PBMCs ($1 \times 10^6$ cells/well) from BLV-uninfected cattle ($n$ = 7 or 8) were incubated with 2 μg/ml 17β-Estradiol for 24 hours Then, cells were harvested and the expression of *COX2* was quantitated by real-time PCR as described above. In addition, culture supernatants were collected after incubation, and then, PGE$_2$ concentrations were measured by ELISA as described above.

To assess the effects of the inhibition of PGE$_2$ receptors in the presence of estradiol, PBMCs ($1 \times 10^6$ cells/well) from uninfected cattle ($n$ = 10) were pretreated with each EP antagonist (EP1: SC-19220, EP2: AH 6809, EP3: L-798,106, EP4: ONO-AE3-208; Cayman Chemical) for 1 hour. After preincubation, PBMCs were washed, and then, cultured with 2 μg/ml of 17β-Estradiol in the presence of 1 μg/ml anti-CD3 mAb and 1 μg/ml anti-CD28 mAb for 24 hours. After cultivation, culture supernatants were collected, and IFN-γ concentrations were quantitated by ELISA as described above. In the cultures using PBMCs from uninfected cattle, two or three independent PBMC-cultures were performed, and pooled data from these experiments were analyzed by the statistical analysis.

To evaluate BLV-specific Th1 responses in BLV-infected pregnant cattle (animals #1–#5), PBMCs ($4 \times 10^5$ cells/well) from BLV-infected cattle were incubated with 0.1 μg/ml BLV gp51 peptide mix [21] for 6 days. Three culture wells were prepared in each sample. After incubation, IFN-γ concentrations in culture supernatants were quantitated by ELISA, as described above.

## Estradiol administration

To evaluate the effects of exogenous estradiol *in vivo*, estradiol administration was performed using BLV-infected cattle (animals #6–#10, Table 1). 2.5 mg/100 kg of estradiol benzoate (OVAHORMON INJECTION; Aska Animal Health, Tokyo, Japan) was inoculated intramuscularly, and peripheral blood samples were collected from these cattle before inoculation (Day 0) and at 1 and 6 hours and 1, 2, 3, 4, and 7 days after inoculation. PGE$_2$ and estradiol concentrations in serum were measured by ELISA, as described above. Whole-blood culture and PBMC culture were conducted to evaluate BLV-nonspecific and BLV-specific Th1 responses, respectively.

## Statistical analysis

Differences were assessed using the Wilcoxon signed-rank test, Welch's *t* test, and the Steel-Dwass test using the EZR software [22]. A *p* value of less than 0.05 was considered statistically significant.

## Results

### The relationship between parturition and the onset of bovine leukosis

Previous studies have shown that a stressful situation, such as parturition, is considered one of the risk factors for the onset of bovine leukosis [10, 11]. In this study, to examine the relationship between parturition and the onset, we analyzed the onset time of 68 cases which were diagnosed with bovine leukosis. As shown in Fig 1, 28 of the 68 specimens (41.18%) developed bovine leukosis during two months before and after delivery (Fig 1A and 1B). These results suggest the association of parturition with the progression to the clinical stage in BLV-infected cattle, although the detailed mechanisms are still unclear.

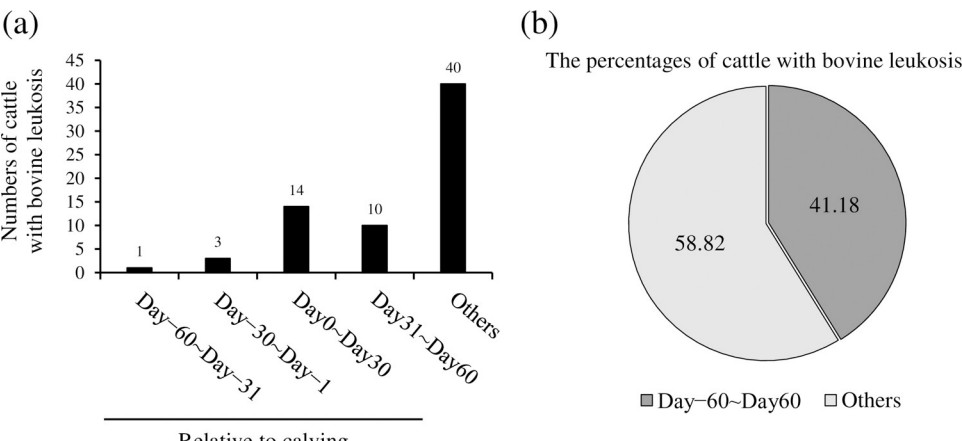

**Fig 1. The relationship between the onset of bovine leukosis and parturition.** The numbers (a) and the percentages (b) of cattle diagnosed as bovine leukosis ($n = 67$). The day of delivery was considered as day 0.

## Inhibition of Th1 responses in BLV-infected pregnant cattle

To reveal the association of parturition with the progression of BLV infection, in this study, we focused on the immunosuppressive function of $PGE_2$, which is also an important factor for the induction of parturition in pregnant animals. To investigate whether $PGE_2$ is involved in immune dysfunction in pregnant cattle, we examined the $PGE_2$ kinetics and Th1 responses of BLV-infected pregnant cattle (Fig 2A–2C and S1 Fig). Compared to the samples collected about 2 weeks before the parturition (Pre, day −17–day −12), $PGE_2$ concentrations in sera were significantly increased on the day of parturition (day 0, Fig 2A). In contrast, Th1 responses against Con A were significantly suppressed on day 0 (Fig 2B). Th1 responses against BLV antigen tended to be suppressed on day 0, although differences were not statistically significant ($p = 0.086$, Fig 2C). Taken together, these results suggest that $PGE_2$ induction before delivery might associated with the inhibition of Th1 responses in pregnant cattle.

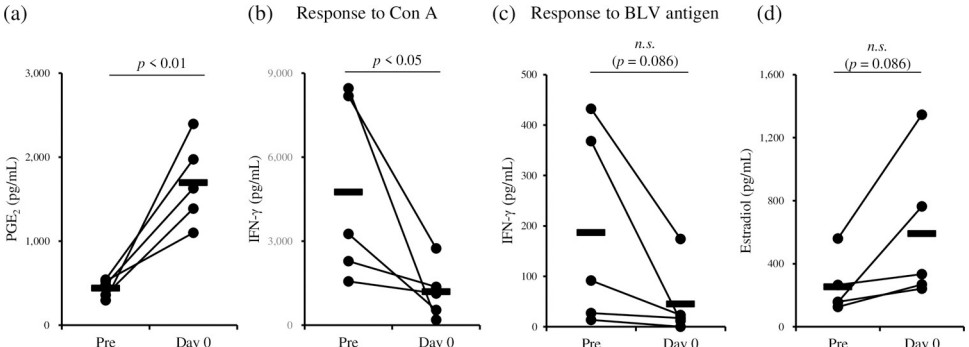

**Fig 2. Analyses of BLV-infected pregnant cattle ($n = 5$, animals #1–#5).** (a) The concentrations of $PGE_2$ in the sera were determined by ELISA. (b and c) Whole-blood cultures or PBMC cultures were performed to evaluate IFN-γ production in response to Con A or gp51 peptide mix, respectively. (d) The concentrations of estradiol in the sera were determined by ELISA. (a–d) Statistical significance was determined by Welch's $t$ test. Pre: day −17 for animal #1, day −16 for animals #2 and #5, day −14 for animal #3, and day −12 for animal #4. Day 0: the day of delivery, $n.s.$: not significance.

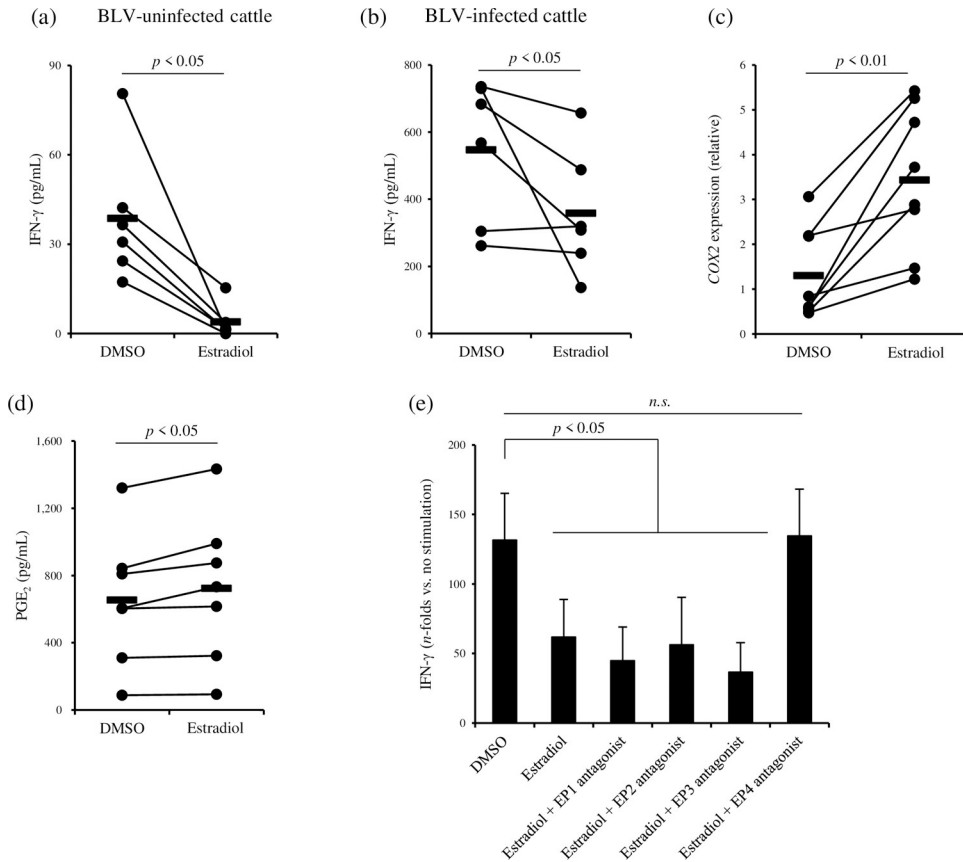

**Fig 3. Immunosuppressive function of estradiol via PGE$_2$ induction.** (a and b) PBMCs from BLV-uninfected or BLV-infected cattle were cultured with estradiol in the presence of anti-CD3 mAb and anti-CD28 mAb or FLK-BLV, respectively (a: BLV-uninfected cattle, $n = 6$, b: BLV-infected cattle, $n = 6$). DMSO was used as a vehicle control of estradiol. IFN-γ concentrations in culture supernatants were quantitated by ELISA. (c and d) PBMCs from BLV-uninfected cattle were cultivated with estradiol, and *COX2* expression (c, $n = 8$) and PGE$_2$ concentration (d, $n = 7$) were measured by real-time PCR and ELISA, respectively. (e) PBMCs from BLV-uninfected cattle ($n = 10$) were pretreated with each EP antagonist for one hour, and then cultured with estradiol in the presence of anti-CD3 mAb and anti-CD28 mAb. IFN-γ production in culture supernatant was determined by ELISA. (a–d) Statistical significances were determined by the Wilcoxon signed-rank test. (e) Data are presented as mean ± SD, and statistical significance was determined by the Steel-Dwass test.

## Immunosuppressive function of estradiol via PGE$_2$ induction *in vitro*

To investigate the mechanism of PGE$_2$ induction in blood before calving, estradiol concentrations in the sera of BLV-infected pregnant cattle were measured using ELISA. Interestingly, in the same manner as PGE$_2$ kinetics in the pregnant cattle, estradiol concentrations tended to be increased on day 0 (Fig 2D and S1 Fig). Previous studies also have shown that the production of estrogen in blood is induced soon before parturition [23–25]. On the basis of these findings, we hypothesized that estradiol plays important roles in the induction of PGE$_2$ production, which causes the suppression of Th1 responses, in pregnant cattle. In order to examine this hypothesis, PBMCs from BLV-uninfected cattle were cultivated with estradiol in the presence of T-cell stimulation. IFN-γ concentrations in culture supernatants were reduced by the treatment of estradiol (Fig 3A). Additionally, estradiol impaired IFN-γ production from PBMCs of BLV-infected cattle in the presence of BLV antigen (Fig 3B). To assess whether estradiol induces PGE$_2$ production, PBMCs from BLV-uninfected cattle were cultivated with estradiol. After incubation, both *COX2* expression and PGE$_2$ production were significantly increased by

treatment with estradiol (Fig 3C and 3D). Furthermore, PBMCs from BLV-uninfected cattle were pretreated with each EP antagonist, and then cultured with estradiol in the presence of T-cell stimulation. The suppressive effect of estradiol was not observed when estradiol incubation was performed in combination with EP4 antagonist (Fig 3E), but not with antagonists of other EP receptors, thus suggesting that estradiol inhibits Th1 responses via $PGE_2$/EP4 signaling.

### Estradiol-induced impairment of Th1 responses *in vivo*

To assess the immunosuppressive function of estradiol *in vivo*, the administration of estradiol was performed using 5 BLV-infected cattle (animals #6–#10, Fig 4 and S2 Fig). We confirmed that the concentrations of estradiol in sera were increased by estradiol administration (Fig 4A). $PGE_2$ concentrations in sera from these infected cattle were significantly increased for 3 days after estradiol administration (Fig 4B). By contrast, IFN-γ production from whole blood cells stimulated with Con A was suppressed by estradiol administration (Fig 4C). Additionally, as shown in Fig 4D, BLV-specific Th1 responses were also impaired by estradiol administration. These results suggest that estradiol inhibits BLV-specific Th1 responses via $PGE_2$ production *in vivo*.

## Discussion

$PGE_2$ is known as an important factor that is related to the induction of parturition in cattle [18]. Additionally, our previous studies have shown that $PGE_2$ has suppressive effects on immune responses, particularly those involving Th1 [16, 17, 26]. However, the association of $PGE_2$ induction during parturition with the suppression of immune responses is still unknown. Therefore, in the current study, we examined whether $PGE_2$ was associated with the suppression of immune responses in periparturient period. Our data showed that $PGE_2$ was induced, and that Th1 responses were inhibited before parturition in pregnant cattle. In addition, both BLV-specific and BLV-nonspecific Th1 responses were suppressed before parturition in pregnant cattle infected with BLV. During BLV infection, the suppression of Th1 responses against BLV antigen is associated with disease progression [27, 28]. In the advanced stages of BLV infection, BLV-specific Th1 responses are downregulated, thus leading to further disease progression and possible EBL [29]. Thus, $PGE_2$ mediated inhibition of BLV-specific Th1 responses around parturition might be a mechanism underlying progression to the clinical stage of BLV infection. In this study, no significant change of BLV proviral load was observed in periparturient period in BLV-infected pregnant cattle (data not shown). However,

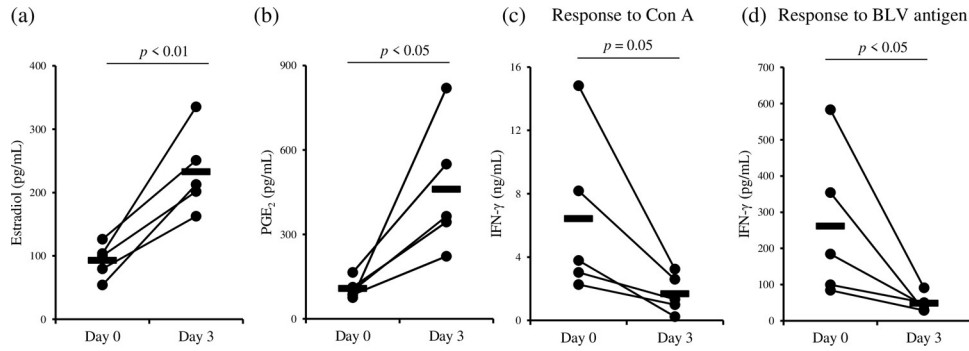

**Fig 4. Functional analysis of estradiol using BLV-infected cattle (*n* = 5, animals #6–#10).** (a) Serum estradiol concentrations were measured by ELISA. (b) Serum $PGE_2$ concentrations were measured by ELISA. (c and d) Whole-blood cultures or PBMC cultures were performed to evaluate IFN-γ production in response to Con A or gp51 peptide mix, respectively. (a–d) Statistical significance was determined by Welch's *t* test (a–d).

previous reports have shown that $PGE_2$/EP4 signaling facilitates BLV viral gene transcription via the activation of cyclic-AMP/protein kinase A/cAMP-response element signaling [17, 30]. Thus, additional experiments with larger numbers of BLV-infected pregnant cattle are needed to examine the effect of $PGE_2$ on BLV proviral load during parturition.

In this study, we identified a novel mechanism of $PGE_2$ upregulation via estradiol. Previous studies have shown that treatment with estradiol induces NF-κB activation in human and rat models [31, 32]; moreover, NF-κB activation is essential for COX-2 induction [14, 33]. Therefore, estradiol presumably induces $PGE_2$ production in the blood via NF-κB activation. In addition, our data showed that $PGE_2$/EP4 signaling involved at least in the suppressive effect of estradiol. By contrast, we did not observe a significant difference between the group treated with estradiol alone and the group treated with estradiol after pretreatment with the EP2 antagonist. Generally, EP2 and EP4 are receptors, which are associated with the immunosuppressive activity of $PGE_2$. However, EP4 represents a high-affinity receptor, whereas EP2 requires significantly higher concentrations of $PGE_2$ for effective signaling [15]. Therefore, the differential effects of EP2 and EP4 blockades are presumably due to the affinity of each receptor.

Tregs are critical for maternal tolerance [4]. Several reports have demonstrated that estrogen drives the induction of Tregs in murine models [34, 35]. Additionally, our previous reports have shown that $PGE_2$ upregulates the expression levels of *Foxp3* and *TGFβ1* in bovine PBMCs [17], and suppressive function of Tregs is associated with the disease progression during BLV infection [36]. Collectively, the available data suggest that during parturition, Tregs induced by the estradiol/$PGE_2$ pathway might suppress Th1 responses and contribute to the progression of BLV infection. Further studies are warranted to investigate the involvement of Tregs in the suppression of Th1 responses during parturition.

Our previous studies have shown that $PGE_2$ upregulation is also involved in the progression of other bovine chronic diseases, such as *Mycoplasma bovis* infection and Johne's disease, which is caused by *Mycobacterium avium* subsp. *paratuberculosis* [16, 25]. Additionally, in terms of Johne's disease, it has been reported that the stress related to parturition is a risk factor for eliciting the onset of the clinical stage of the disease [37]. However, no report has been examined whether $PGE_2$ induction during parturition inhibits *M. bovis*-specific and *M. avium* subsp. *paratuberculosis*-specific Th1 responses. The involvement of the estradiol/$PGE_2$ pathway in Th1 suppression during parturition should be determined in further experiments by using pregnant cattle with these diseases.

In conclusion, during parturition, BLV-specific Th1 responses are suppressed via estradiol-induced $PGE_2$, which contributes to the disease progression of BLV-infection. Additionally, estradiol-induced $PGE_2$ also inhibited BLV-nonspecific Th1 responses, thus suggesting that $PGE_2$ might be involved in susceptibility to opportunistic infections in the periparturient period. To the best of our knowledge, this is the first study to show the suppressive function of the estradiol/$PGE_2$/EP4 pathway. Further studies will open up new avenues for the control of infections during parturition both in pregnant cattle and in pregnant women.

## Supporting information

**S1 Checklist. The ARRIVE guidelines 2.0: Author checklist.**
(PDF)

**S1 Fig. Analyses of BLV-infected pregnant cattle.** (a) The concentrations of $PGE_2$ in the sera were determined by ELISA. (b and c) Whole-blood culture or PBMC culture was performed to evaluate IFN-γ production in response to Con A or gp51 peptide mix, respectively. (d) The concentrations of estradiol in the sera were determined by ELISA.
(PPTX)

**S2 Fig. Functional analysis of estradiol using BLV-infected cattle.** (a–d) Estradiol administration was performed using BLV-infected cattle (animals #6–#10). (a) Serum estradiol concentrations were measured by ELISA. (b) Serum PGE2 concentrations were measured by ELISA. (c and d) Whole-blood culture or PBMC culture was performed to evaluate IFN-γ production in response to Con A or gp51 peptide mix, respectively.
(PPTX)

**S1 Table. The numbers of cattle diagnosed as bovine leukosis, related to Fig 1.**
(PPTX)

**S2 Table. The original data of analyses of BLV-infected pregnant cattle (animals #1–#5), related to Fig 2.** (a) The concentrations of $PGE_2$ in the sera. (b and c) IFN-γ production in response to Con A (b) or gp51 peptide mix (c) in the whole-blood cultures (b) or PBMC cultures (c). (d) The concentrations of estradiol in the sera.
(PPTX)

**S3 Table. The original data of immunosuppressive assay using estradiol, related to Fig 3.** (a and b) IFN-γ production in the cultures of PBMCs from BLV-uninfected (a) and infected cattle (b) cultivated with anti-CD3 and anti-CD28 mAbs (a) and FLK-BLV (b), respectively. (c and d) *COX2* expression (c) and $PGE_2$ concentration (d) in PBMCs from BLV-uninfected cattle cultivated with estradiol. (e) IFN-γ production in PBMCs from BLV-uninfected cattle cultivated with each EP antagonist, estradiol, and anti-CD3 and anti-CD28 mAbs.
(PPTX)

**S4 Table. The original data of functional analysis of estradiol using BLV-infected cattle (animals #6–#10), related to Fig 4.** (a) The concentrations of estradiol in the sera. (b) The concentrations of $PGE_2$ in the sera. (c and d) IFN-γ production in response to Con A (c) or gp51 peptide mix (d) in the whole-blood cultures (c) or PBMC cultures (d).
(PPTX)

## Acknowledgments

We are grateful to Dr. Hideyuki Takahashi, Dr. Tomio Ibayashi, and Dr. Yasuyuki Mori for valuable advice and discussions. We thank Enago (www.enago.jp) for the English language review.

## Author Contributions

**Conceptualization:** Yamato Sajiki, Satoru Konnai, Tomohiro Okagawa, Naoya Maekawa, Shiro Murata, Kazuhiko Ohashi.

**Data curation:** Yamato Sajiki, Satoru Konnai, Tomohiro Okagawa, Naoya Maekawa, Shinya Goto, Junko Kohara, Atsushi Nitanai, Hirofumi Takahashi, Kentaro Kubota, Hiroshi Takeda, Shiro Murata, Kazuhiko Ohashi.

**Formal analysis:** Yamato Sajiki, Satoru Konnai, Tomohiro Okagawa, Naoya Maekawa, Shinya Goto, Junko Kohara, Hirofumi Takahashi, Kentaro Kubota, Hiroshi Takeda, Shiro Murata, Kazuhiko Ohashi.

**Funding acquisition:** Satoru Konnai.

**Investigation:** Yamato Sajiki, Satoru Konnai, Tomohiro Okagawa, Shinya Goto, Junko Kohara, Atsushi Nitanai, Hirofumi Takahashi, Kentaro Kubota, Hiroshi Takeda.

**Methodology:** Yamato Sajiki, Satoru Konnai, Tomohiro Okagawa, Junko Kohara, Atsushi Nitanai, Hirofumi Takahashi, Kentaro Kubota, Hiroshi Takeda.

**Project administration:** Satoru Konnai.

**Resources:** Satoru Konnai.

**Supervision:** Satoru Konnai.

**Validation:** Satoru Konnai.

**Visualization:** Satoru Konnai.

**Writing – original draft:** Yamato Sajiki, Satoru Konnai, Naoya Maekawa, Shiro Murata, Kazuhiko Ohashi.

**Writing – review & editing:** Shiro Murata, Kazuhiko Ohashi.

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
