## [Decision Letter · Decision Letter 0]

2 Dec 2021

PONE-D-21-17024

Estradiol - induced immune suppression via prostaglandin E2 during parturition in bovine leukemia virus-infected cattle

PLOS ONE

Dear Dr. Konnai,

Thank you for submitting your manuscript to PLOS ONE. After careful consideration, we feel that it has merit but does not fully meet PLOS ONE’s publication criteria as it currently stands. Therefore, we invite you to submit a revised version of the manuscript that addresses the points raised during the review process.

We look forward to receiving your revised manuscript.

Kind regards,

Juan J Loor

Academic Editor

PLOS ONE

Journal Requirements:

2. As part of your revision, please complete and submit a copy of the Full ARRIVE 2.0 Guidelines checklist, a document that aims to improve experimental reporting and reproducibility of animal studies for purposes of post-publication data analysis and reproducibility: https://arriveguidelines.org/sites/arrive/files/Author%20Checklist%20-%20Full.pdf (PDF). Please include your completed checklist as a Supporting Information file. Note that if your paper is accepted for publication, this checklist will be published as part of your article.

3. You indicated that you had ethical approval for your study. In your Methods section, please ensure you have also stated whether you obtained consent from owners of the cattle included in the study or whether the research ethics committee or IRB specifically waived the need for their consent.

Reviewers' comments:

Reviewer's Responses to Questions

**Comments to the Author**

1. Is the manuscript technically sound, and do the data support the conclusions?

Reviewer #1: Partly

2. Has the statistical analysis been performed appropriately and rigorously? 

Reviewer #1: Yes

3. Have the authors made all data underlying the findings in their manuscript fully available?

Reviewer #1: Yes

4. Is the manuscript presented in an intelligible fashion and written in standard English?

Reviewer #1: No

5. Review Comments to the Author

Reviewer #1: This manuscript addresses a very important need for better definition of factors that impact immunosuppression in the peripartum period. The focus on PGE2 and estradiol is an important consideration and has yielded insight on specific receptor involvement.However, it is unclear if the focus is on at the time of parturition or the surrounding period?. The outcomes that determined specific conclusion on the selected term eg effect at parturition for the observed effects are not clear. More information on age, stage of lactation, parity, breed, diet and management etc would help uncover other possible contributing factors and should at least be discussed. Furthermore, the possible contribution of declining progesterone concentration (known immunosuppressant) and health parameters such status of infection by other pathogens, occurrence of diseases such as laminitis, mastitis etc. are not considered. In light of the impact of LPS on in vitro COX2 expression and PGE2 secretion what were the levels of endotoxins in reagents and containers for in vitro studies and what controls were used to evaluate the possible contribution of LPS? Discuss the use of the DMSO control versus buffer or known plasma. Please check sentence structure/missing words etc. eg line 2 and line 12.1, Not clear what this sentence means please edit Line 2 In dairy cattle, efficient milk production continues to require for pregnancy and 3 parturition each year.

6. PLOS authors have the option to publish the peer review history of their article (what does this mean?). If published, this will include your full peer review and any attached files.

Reviewer #1: **Yes: **Mulumebet Worku

---

## [Author Response · Author response to Decision Letter 0]

28 Dec 2021

Answers to the comments of the Reviewer #1

 Thank you very much for your comments. We appreciate your comments and amended our manuscript accordingly.

 Answers to the comments are as follows:

Reviewer#1s' comments:

This manuscript addresses a very important need for better definition of factors that impact immunosuppression in the peripartum period. The focus on PGE2 and estradiol is an important consideration and has yielded insight on specific receptor involvement. However, it is unclear if the focus is on at the time of parturition or the surrounding period? The outcomes that determined specific conclusion on the selected term eg effect at parturition for the observed effects are not clear. More information on age, stage of lactation, parity, breed, diet and management etc would help uncover other possible contributing factors and should at least be discussed. Furthermore, the possible contribution of declining progesterone concentration (known immunosuppressant) and health parameters such status of infection by other pathogens, occurrence of diseases such as laminitis, mastitis etc. are not considered. In light of the impact of LPS on in vitro COX2 expression and PGE2 secretion what were the levels of endotoxins in reagents and containers for in vitro studies and what controls were used to evaluate the possible contribution of LPS? Discuss the use of the DMSO control versus buffer or known plasma. Please check sentence structure/missing words etc. eg line 2 and line 12.1, Not clear what this sentence means please edit Line 2 In dairy cattle, efficient milk production continues to require for pregnancy and 3 parturition each year.

1. ‘It is unclear if the focus is on at the time of parturition or the surrounding period?. The outcomes that determined specific conclusion on the selected term eg effect at parturition for the observed effects are not clear’

Ans: Immune suppression during pregnancy and parturition is regarded as a risk factor related to the progression of bovine chronic infections, such as bovine leukemia virus (BLV) infection. In our previous study, we demonstrated that prostaglandin E2 (PGE2) suppresses BLV-specific Th1 responses and contributes to the progression of BLV infection (Sajiki et al., J. Immunol., 2019, 203(5):1313-1324.). Thus, we focused on parturition because PGE2 and estradiol, which suppress immunity, are secreted by the placenta. This study was performed to investigate the involvement of estradiol-induced PGE2 in immune suppression during parturition. We believe that our study makes a significant contribution to the literature as a translational medical research because our data suggest that PGE2 upregulation inhibits Th1 responses during parturition. Additionally, estradiol was essential for PGE2 induction in pregnant cattle. Taken together, our data suggest that, in pregnant cattle, estradiol-induced PGE2 is involved in the suppression of Th1 responses, which contributes to the progression of bovine chronic infections.

2. ‘More information on age, stage of lactation, parity, breed, diet and management etc would help uncover other possible contributing factors and should at least be discussed’

Ans: As you pointed out, other factors such as lactation stage, parity, diet and management are also important issues as risk factors for developing leukemia. However, unfortunately, due to various restrictions, the information was not available. In this study, we focused only on parturition because PGE2 and estradiol, which suppress immunity, are secreted by the placenta, but we would like to consider other risk factors in the future. We would like to present it as a follow-up report.

3. ‘the possible contribution of declining progesterone concentration (known immunosuppressant) and health parameters such status of infection by other pathogens, occurrence of diseases such as laminitis, mastitis etc. are not considered’

Ans: Thank you very much for your interesting comments. This study does not show a causal relationship between the onset of the disease and other health parameters such as infection status by other pathogens or the occurrence of diseases such as laminitis and mastitis, but we believe this is something that we should also focus on. Actually, we are currently investigating these points for implementation, and this is one of our future studies. We would like to present it as a follow-up report.

4. ‘In light of the impact of LPS on in vitro COX2 expression and PGE2 secretion what were the levels of endotoxins in reagents and containers for in vitro studies and what controls were used to evaluate the possible contribution of LPS?’

Ans: As you pointed out, LPS is an important factor. To avoid the effects of LPS contamination, we used commercially available endotoxin (LPS)-free regents and containers for this experiment. Therefore, we did not perform any measurements.

5. ‘Discuss the use of the DMSO control versus buffer or known plasma’

Ans: PGE2 and estradiol are extremely insoluble in water. Therefore, DMSO is used as the best solvent. In this study, DMSO was used as the solvent as in our previous experiments. Since high concentrations of DMSO are cytotoxic, we used the same concentration of DMSO as a negative control.

6. ‘Please check sentence structure/missing words etc. eg line 2 and line 12.1, Not clear what this sentence means please edit Line 2 In dairy cattle, efficient milk production continues to require for pregnancy and 3 parturition each year’

Ans: Following your comment, we modified the sentence to ‘In dairy cattle, efficient milk production continues to require for pregnancy and 3 parturition each year’.

---

## [Decision Letter · Decision Letter 1]

25 Jan 2022

Estradiol - induced immune suppression via prostaglandin E2 during parturition in bovine leukemia virus-infected cattle

PONE-D-21-17024R1

Dear Dr. Konnai,

We’re pleased to inform you that your manuscript has been judged scientifically suitable for publication and will be formally accepted for publication once it meets all outstanding technical requirements.

Kind regards,

Juan J Loor

Academic Editor

PLOS ONE

Additional Editor Comments (optional):

Reviewers' comments:

Reviewer's Responses to Questions

**Comments to the Author**

1. If the authors have adequately addressed your comments raised in a previous round of review and you feel that this manuscript is now acceptable for publication, you may indicate that here to bypass the “Comments to the Author” section, enter your conflict of interest statement in the “Confidential to Editor” section, and submit your "Accept" recommendation.

Reviewer #1: All comments have been addressed

2. Is the manuscript technically sound, and do the data support the conclusions?

Reviewer #1: Yes

3. Has the statistical analysis been performed appropriately and rigorously? 

Reviewer #1: Yes

4. Have the authors made all data underlying the findings in their manuscript fully available?

Reviewer #1: Yes

5. Is the manuscript presented in an intelligible fashion and written in standard English?

Reviewer #1: Yes

6. Review Comments to the Author

Reviewer #1: Thank you for your response. I would like to encourage follow up studies to uncover contributing factors for improved management decsions and translational applications.

7. PLOS authors have the option to publish the peer review history of their article (what does this mean?). If published, this will include your full peer review and any attached files.

Reviewer #1: **Yes: **Mulumebet Worku

---

## [Editor Report · Acceptance letter]

14 Feb 2022

PONE-D-21-17024R1 

Estradiol-induced immune suppression via prostaglandin E2 during parturition in bovine leukemia virus-infected cattle 

Dear Dr. Konnai:

I'm pleased to inform you that your manuscript has been deemed suitable for publication in PLOS ONE. Congratulations! Your manuscript is now with our production department. 

Kind regards, 

on behalf of

Dr. Juan J Loor 

Academic Editor

PLOS ONE